# Impact of Pipe Material and Temperature on Drinking Water Microbiome and Prevalence of *Legionella*, *Mycobacterium*, and *Pseudomonas* Species

**DOI:** 10.3390/microorganisms11020352

**Published:** 2023-01-31

**Authors:** Saleh Aloraini, Absar Alum, Morteza Abbaszadegan

**Affiliations:** 1Department of Civil Engineering, College of Engineering, Qassim University, Buraydah 52571, Saudi Arabia; 2School of Sustainable Engineering and the Built Environment, Arizona State University, Tempe, AZ 85281, USA; 3Water and Environmental Technology Center, Arizona State University, Tempe, AZ 85281, USA

**Keywords:** drinking water microbiome, pipe material, water temperature, *Legionella*, *Pseudomonas*, *Mycobacterium*, differential abundance analysis

## Abstract

In drinking water distribution systems (DWDSs), pipe material and water temperature are some of the critical factors affecting the microbial flora of water. Six model DWDSs consisting of three pipe materials (galvanized steel, copper, and PEX) were constructed. The temperature in three systems was maintained at 22 °C and the other 3 at 32 °C to study microbial and elemental contaminants in a 6-week survey using 16S rRNA next-generation sequencing (NGS) and inductively coupled plasma-optical emission spectrometry (ICP-OES). Pipe material and temperature were preferentially linked with the composition of trace elements and the microbiome of the DWDSs, respectively. *Proteobacteria* was the most dominant phylum across all water samples ranging from 60.9% to 91.1%. Species richness (alpha diversity) ranking was PEX < steel ≤ copper system and elevated temperature resulted in decreased alpha diversity. *Legionellaceae* were omni-prevalent, while *Mycobacteriaceae* were more prevalent at 32 °C (100% vs. 58.6%) and *Pseudomonadaceae* at 22 °C (53.3% vs. 62.9%). Heterogeneity between communities was disproportionately driven by the pipe material and water temperature. The elevated temperature resulted in well-defined microbial clusters (high pseudo-F index) in all systems, with the highest impact in PEX (10.928) followed by copper (9.696) and steel (5.448). *Legionellaceae* and *Mycobacteriaceae* are preferentially prevalent in warmer waters. The results suggest that the water temperature has a higher magnitude of impact on the microbiome than the pipe material.

## 1. Introduction

Globally, drinking water distribution systems (DWDSs) represent one of the largest manmade networks. In the U.S., it spans a collective distance of more than 2 million miles delivering more than 39 million gallons per day (MGD) of drinking water to end users [1]. These systems are far from being sterile and several disinfectants (usually chlorine or chloramine) are commonly used to suppress bacterial growth and to biologically stabilize DWDSs. However, the bacterial counts in the nutrient-poor municipal treated water are abundant ranging from 10^3^ to 10^6^ cells per mL [2]. 

These networks sustain bacterial ecology (both planktonic and sessile) characterized for diversity which is affected by many factors such as source water, temperature, pipe material, water chemistry, and other factors. Multiple studies have shown that opportunistic pathogens (OPs) such as *Legionella pneumophila*, *Mycobacterium avium*, and *Pseudomonas aeruginosa* can proliferate and enteric viruses and parasitic protozoa can survive in biofilms [3]. It is estimated that 95% of the total bacteria are in biofilms and only 5% are planktonic, which are usually sampled while monitoring for water quality assessment [3]. 

Treated water travels long distances in main distribution systems and in-premise plumbing to reach the end user. The DWDSs environment, with variations in water temperature, chlorine concentration, and water residence time, creates niches favorable for bacterial growth and survival [4,5]. 

It has also been suggested that the proliferation of commensal bacterial and opportunistic pathogens in premise plumbing may occur under conditions where water is warmer and stagnant resulting in a faster decay of disinfection residuals [4,5]. These conditions are usually observed within premise plumbing right before the delivery of water to taps [6,7]. Multiple studies have reported an association between the colonization of *Legionella* spp. in DWDSs with the presence of trace metals such as Cu, Mn, and Zn [8,9]. Additionally, *Mycobacteria* spp. abundance has been associated with the presence of assimilable organic carbon (AOC) concentration [10]. Microbial water quality can be influenced by a variety of factors, including the type of pipe material, the presence of nutrients, and disinfection practices. The age and type of pipe material have been found to influence the rate of biofilm formation where new copper pipe (<200 days old) showed slower biofilm formation compared to polyethylene pipes, but that difference becomes less significant after 200 days [11]. The deterioration of water quality in the premise plumbing is simulated by the maturity of biofilm formation, suspended cell counts, and relevant water parameters [11]. Hot water systems at certain temperatures and flow rates tend to have higher microbial counts when compared to cold water systems [12].

In this study, the impact of pipe materials and temperature on six model DWDS microbiomes was investigated. In addition, a probiotic approach to control bacterial growth in three different DWDS pipe materials and two water temperatures on the relative abundance of the OPs was investigated. This approach may provide relevant data in the knowledge gap in the resilience and proliferation of bacterial pathogens in DWDSs. 

## 2. Materials and Methods

### 2.1. System Design and Sample Collection Procedure

Six pilot-scale model drinking water distribution systems (DWDSs) were constructed using three pipe materials including PEX, copper, and galvanized steel (Figure 1). The overall length of the 3/4-inch inner diameter pipe from exit to reentry to the reservoir was 144 inches, with 72 inches on the shelf. Each DWDS had a 10-gallon polyethylene tank serving as a reservoir. Systems were operated at a volumetric flow rate of 15 L/min (4 GPM) with a linear velocity of 0.77 m/s. For each pipe material, two identical systems were used to simultaneously study the impact of pipe material and water temperature on the drinking water microbiome over a 6-week period. The temperature in three systems was maintained at 32 °C and the other three systems were operated as control at ambient temperature, 22 °C. The 32 °C temperature was maintained using 100-watt aquarium heaters (Aqueon, Franklin, WI, USA) that gradually increased the temperature without overheating in 24 h. The water temperature was regularly monitored using a dual infrared thermometer (cat# E650d ennoLogic, Eugene, OR, USA). The average temperature of the heated and non-heated systems during the study was 32 ± 0.5 °C and 22 ± 0.5 °C, respectively. For microbial analyses, 500 mL of water samples were collected twice a week (Wednesdays and Saturdays) and after each sample collection, 33% of the total water in each system was replenished with fresh tap water (City of Tempe, AZ, USA) to maintain reasonable hydraulic retention time in the systems. For metal, 50 mL water samples were collected in glass vials and acid-preserved with 2% by volume of 0.32 M of HNO_3_ to prevent metal precipitation.

### 2.2. Chemical Analysis of Water Samples

For trace elements, 50 mL water samples were filtered using 0.22 µM disk membranes (MilliporeSigma™ SLHVR33RS, Burlington, MA, USA) to remove their microbial content, and then the samples were analyzed using inductively coupled plasma optical emission spectrometry (ICP-OES) (Agilent 5900 Synchronous Vertical Dual View (SVDV), Santa Clara, CA, USA) following manufacturer protocols. The metal compositions of the water samples were determined according to the US EPA 200.7 method [13]. A total of 26 trace elements were measured and the 11 trace elements (Na, Li, As, Ca, Cu, Pb, Al, Ni, Mg, K, and Zn) that were either significantly or differently present, were further analyzed. Relationships between the concentrations of these trace elements and the type of DWDSs were determined by principal component analysis (PCA) correlation. PCA ordination biplot was generated using XLSTAT [14] (Figure 2). 

### 2.3. Sample Processing and DNA Extraction

Throughout the study, a total of 500 samples were collected in sterile plastic bottles. For each system, 500 mL of water was filtered using 47 mm mixed cellulose ester filters with a pore size of 0.22 µM (GSWP047S0, EMD Millipore). Microbial DNA was extracted using the DNeasy PowerSoil Kit (QIAGEN, Hilden, Germany). Briefly, the filters were aseptically cut into small pieces and added to the PowerSoil bead tube and proceeded to DNA extraction according to the manufacturer’s instruction.

### 2.4. 16S Illumina Sequencing

Libraries of 16S were generated using next-generation sequencing in the MiSeq Illumina platform. Amplicon sequencing of the V4 region of the 16S rRNA gene was generated with a barcoded primer set 515f/806r designed by Caporaso et al. [15] following the Earth Microbiome Project protocol (EMP) (http://www.earthmicrobiome.org/emp-standard-protocols/ (accessed on 19 September 2022) for the library preparation. PCR amplification was performed in duplicate, pooled, and quantified using AccuBlue^®^ dsDNA Quantitation Kit. Negative control (no template sample in library preparation) was included in DNA library preparation to confirm the absence of extraneous DNA contamination. A total of 240 ng of DNA per sample was pooled in a 1.5 mL Eppendorf tube and then cleaned using QIA quick PCR purification kit (QIAGEN). The DNA pool was quantified using Qubit Fluorometer and a qPCR was performed using the NEBNext Library Quant Kit for Illumina (New England Biolabs) following the manufacturer’s instructions. The pooled DNA was diluted to a final concentration of 4 nM then denatured and diluted to a final concentration of 4 pM with a 25% PhiX. Finally, the DNA library was loaded in the MiSeq Illumina and run using the version 2 module, 2 × 250 paired-end, following the directions of the manufacturer.

### 2.5. Microbiome Statistical Analysis

Microbiome Analysis was performed using QIIME2 (2022.08 version). Briefly, the paired-end sequences were imported and demultiplexed into QIIME 2 artifact [16]. The paired-end demultiplexed sequence was then denoised, dereplicated, and filtered out from chimers using the DADA2 pipeline with standard truncation and trimming of low-quality ends [17]. For the 66 samples, a total of 1513 features with a total frequency of 1,566,315 were obtained. Samples with less than 18,760 reads, features that represent Mitochondria and chloroplast, and features with very low abundance (less than 10 frequencies across all samples) were removed using QIIME2. This resulted in an OTU table containing 61 samples with 555 features and a total frequency of 1,466,314. The OTUs were then classified by taxon using classify-sklearn trained with the GreenGenes ribosomal RNA gene database (13_8 99% OTUs) [18]. After the classification, the detection frequency of the selected OPs was calculated at the family taxon level (Table 1). For bacteria with most abundant class, the mean relative abundance was calculated for each system (Figure 3).

Rooted and unrooted phylogenetic trees for alpha and beta diversity analysis were created using Fasttree and Mafft alignment [19].The alpha diversity metrics (Shannon entropy, Pielou’s evenness, and Faith’s phylogenetic diversity (Faith’s PD)) and beta diversity metrics were performed using QIIME 2 pipeline with sample depth rarefaction of 18,670 frequency (Figure 4). The change over time on the alpha diversity based on the number of observed features was calculated using QIIME 2 lg in (q2-longitudinal) (Figure 5).Similarity and dissimilarity of the microbiome samples were computed using phylogenetic and non-phylogenetic-based beta diversity metrics. The phylogenetic-based beta diversity metrics were weighted UniFrac and Unweighted UniFrac. While non-phylogenetic-based beta diversity metrics were Bray–Curtis dissimilarity and Jaccard similarity index (Figure 6). Additionally, using unweighted UniFrac distances, pairwise comparisons of significance in differences within the microbial composition based on pipe material and temperature condition were assessed using PERMANOVA (permutational multivariate analysis of variance) with 999 permaturations per test (Table 2) as described in Anderson [20]. 

Differential abundance analysis, which aims to identify the significant differences in the abundance of features between the 32 °C and 22 °C DWDSs was performed by ALDEx2 [18] as QIIME2 plug-in. ALDEx2 was used to calculate the differential relative abundances between the 32 °C and 22 °C systems. The tool also calculates the sampling error using a Dirichlet-multinomial model to estimate abundances from counts. The sampling and biological variation can then be inferred by taking a Monte Carlo approach to calculate the expected false discovery rate (FDR) and the Benjamini–Hochberg corrected *p*-value using Welch’s *t*-test. The proportions resulting from Monte Carlo instances are transformed using the centered log-ratio (clr) transformation. The clr abundance values of each feature represent its relative abundance to the mean of all abundant features in the sample. Features with FDR of less than 0.05 are considered significantly differentially abundant, and their different abundances are not due to random sampling error [21]. In the present study, we assumed that features that have ±5 median per-feature difference in (clr) between condition A (32 °C) and B (22 °C) are significantly differentially abundant. A total of 20 taxa (classified at the family level) were identified which were significantly differentially abundant with a maximum *p*-value of 0.0022. The evenness and richness were calculated using 3 diversity metrics: Faith PD, Pielou’s evenness, and Shannon entropy. 

To predict the metagenomic functional content of the model DWDSs with different pipe material and temperature conditions, PICRUST (Phylogenetic Investigation of Communities by Reconstruction of Unobserved States) was used to predict the gene function profile with quantifiable uncertainty of each pipe material system using only a marker gene such as 16S rRNA gene [22]. Briefly, the feature (OTU) table was converted from QIIME2 artifact into a JSON version BIOM file to be compatible with downstream applications outside QIIME2 environment and was then uploaded into Huttenhower’s Lab within the Galaxy web application (https://huttenhower.sph.harvard.edu/galaxy/ accessed on 9 November 2022) [23]. First, the 16S copy number was normalized to better reflect the true abundance of each OTU. Second, the virtual metagenome of KEGG orthologs of each sample was predicted from the normalized 16S rRNA gene number. Finally, the predicted virtual metagenome collapsed to the desired KEGG Pathway hierarchy level for PICRUSt prediction. The metagenome functional profiles were statistically analyzed using STAMPS Software [24]. The extended error bars with mean proportion and 95% confidence interval were used to compare the differential abundance of gene functions in 32 °C and 22 °C DWDSs of different pipe materials (galvanized steel, copper, and PEX).

The temporal trend in the alpha diversity heterogeneity of the model DWDSs was evaluated by conducting a longitudinal analysis of our microbiome data. The QIIME2 plug-in, the q2-longitudinal package (https://github.com/qiime2/q2-longitudinal accessed on 23 September 2022) [25], was used to perform linear mixed effects (LME) models for the systems categorized by independent variables (fixed effects) such as the pipe material and the temperature condition using alpha diversity as the longitudinal response. The analysis was conducted to detect the change in alpha diversity of each system over time. The LME models in the q2-longitudinal package are computed using statsmodels’ “mixedlm” function [26].

## 3. Results

### 3.1. Characterization of Chemical Variables in the Model Distribution Systems

Only 11 out of the 26 trace elements analyzed were found to have concentrations above the limit of detection (LoD). These elements are Na, Li, As, Ca, Cu, Pb, Al, Ni, Mg, K, and Zn, and the concentration of five elements (Na, Li, Ca, K, and Mg) was found to be neither affected by pipe materials nor by temperature. Copper pipe systems had higher concentrations of Cu and Pb, while in PEX and galvanized steel systems Ni and Zn were detected in elevated concentrations. Using PCA ordination biplot of elemental analysis, water samples were better separated based on pipe material than temperature (Figure 2). This is believed to be directly linked to the chemical composition of the pipe material used in each system.

**Figure 2 microorganisms-11-00352-f002:**
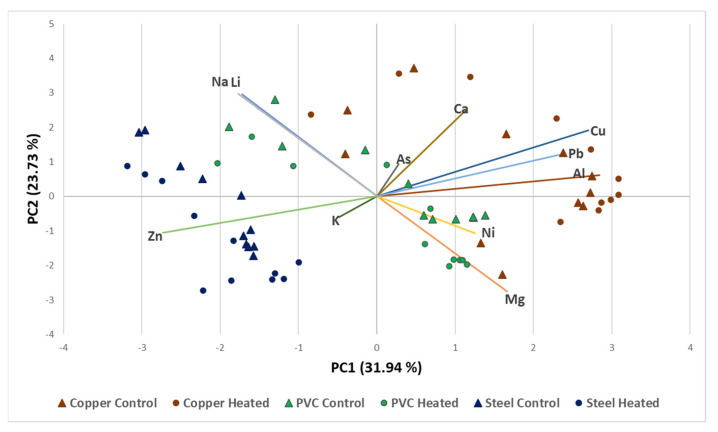
Divergence among DWDSs based on the PCA ordination biplot of chemical analysis.

### 3.2. Bacterial Diversity of the Water Samples in the Model Water Distribution Systems

Bacterial diversity analyses were performed after the removal of *Mitochondria, Chloroplast,* and all the reads with a frequency of less than 10 across all samples. Thereafter, rarefaction was performed to a sequence depth of 18,670 to all samples resulting in the retention of most of the features across the samples (72.71%) while minimizing the impact of uneven sample depth on downstream analysis. The total OTUs remaining after applying filtration and rarefaction were 555 features (61 out of 66 samples) with a frequency that ranged from 10 to 245,079 counts with a median frequency of 111 counts. The taxonomy assignment of the clustered OTUs resulted in a total of 20 phyla containing 44 different classes of bacteria (Figure 3). The taxonomy analysis revealed *Proteobacteria* as the most abundant phylum across all samples with a relative frequency that ranges from 60.9% to 91.1% (average of 80.4%) followed by *Cyanobacteria* (6.7%) and *Bacteroidetes* (4.9%) and *Planctomycetes* (3.4%). In-depth class level analysis identified multiple cases where a specific class of bacteria was prevalent in one of the two temperature conditions in the same pipe material systems. *Nitrospira* (*Nitrospirales*) and *Gemmatimonadetes* are prevalent in the 32 °C PEX system with an average relative frequency of 4.2% and 4.0% compared to 0.4% and 0.06% in the 22 °C PEX system, respectively. *Alphaproteobacteria* (*Caulobacterales*) were one of the most abundant bacteria in the 22 °C PEX system with an average relative frequency of 24.6% compared to 5.9% in the 32 °C PEX system. *Alphaproteobacteria* (*Caulobacterales*) and *Cyanobacteria* (4C0d-2) have a higher average relative abundance in all the systems operated at 22 °C compared to the 32 °C systems. Taxa containing opportunistic pathogens (OPs), *Legionella pneumophila*, *Mycobacterium avium*, *Pseudomonas aeruginosa* taxa were widely detected across all the model DWDSs with a frequency of detection between 100 to 27% (Table 1). *Legionellaceae* were detected in all 66 samples across all the model DWDSs. *Mycobacteriaceae* were detected in 100% of the systems operated at 32 °C and had a lower prevalence frequency in the systems operated at 22 °C. On average, *Pseudomonadaceae* were detected more frequently in the 22 °C compared to the 32 °C systems. *Pseudomonadaceae* detection frequency was the lowest in copper systems and the highest in galvanized steel systems. 

**Table 1 microorganisms-11-00352-t001:** Frequency of detection of OPs on family taxon level across the 6 model DWDSs.

Temperature Condition	Warm			Cold		
Material Type	Copper	PEX	Steel	Copper	PEX	Steel
*Legionellaceae*	100%	100%	100%	100%	100%	100%
*Pseudomonadaceae*	20%	50%	90%	36.4%	63.6%	88.9%
*Mycobacteriaceae*	100%	100%	100%	81.8%	27.3%	66.7%

**Figure 3 microorganisms-11-00352-f003:**
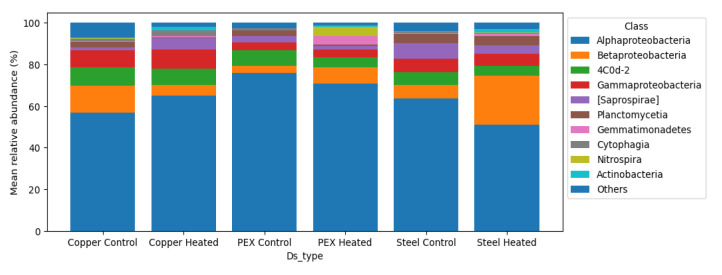
Mean relative abundance of the ten most abundant classes of bacteria in the model DWDSs. (Based on the average relative abundance of all samples for each DWDS).

### 3.3. Species Richness and Evenness for Different Distribution Systems—Alpha Diversity

Alpha diversity was calculated to evaluate the impact of pipe material and temperature on the richness and evenness of bacterial communities. Alpha diversity metrics were computed using three methods: Faith’s PD, Pielou’s evenness, and Shannon entropy methods (Figure 4). The Kruskal–Walli test was used to determine the *p*-values of each metric calculated to be 0.0008, 0.0069, and 0.0011 for Faith’s PD, Pielou’s evenness, and Shannon entropy, respectively. Based on *p*-values, the difference in the bacterial diversity of the six systems constructed with different pipe materials and operated at two temperatures is considered statistically significant (*p* < 0.05). The Faith’s PD of a sample is calculated using the sum of the minimum branch lengths connecting the phylogenetic tree of the microbial community. Alpha diversity measures using Faith’s PD show that copper and PEX systems at 22 °C have slightly higher median phylogenetic diversity values compared to the systems at 32 °C. In contrast, the median Faith’s PD for the galvanized steel system at 32 °C is higher than the 22 °C system. The alpha diversity based on Pielou’s evenness and Shannon entropy showed a slight increase in species richness and evenness for all three 32 °C systems compared to the 22 °C systems. 

**Figure 4 microorganisms-11-00352-f004:**
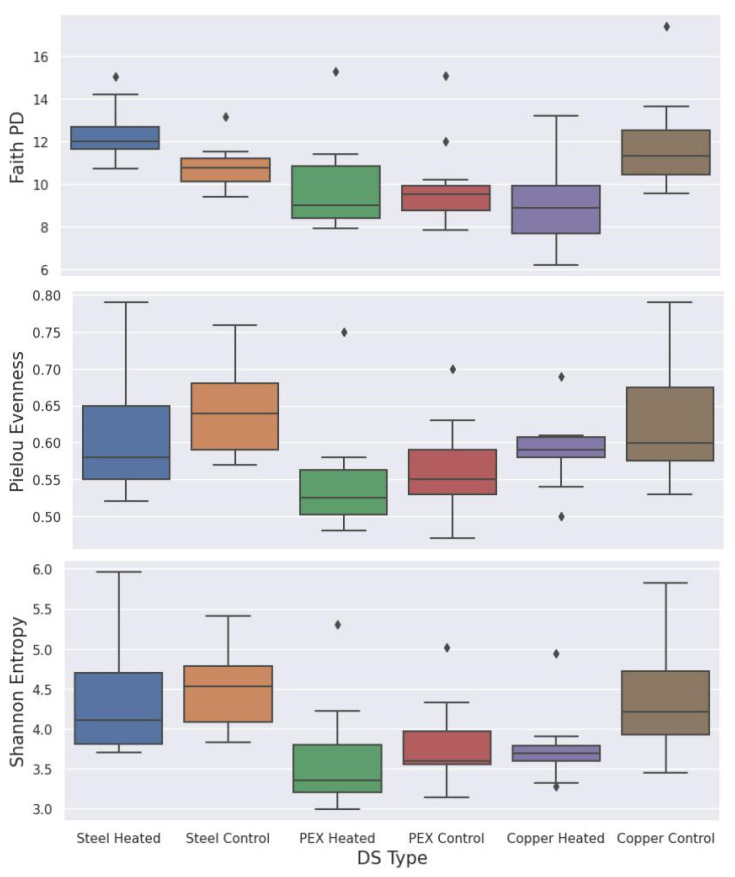
Alpha diversity based on Faith’s PD, Pielou’s evenness, and Shannon entropy metrics.

In addition, the LME model was used to track the temporal trends in alpha diversity (number of species) for each variable in the model DWDSs (Figure 5A). The impact of pipe material and temperature conditions on the number of bacterial species in each system indicate a steady decline between 57.5% to 23.5% from their highest numbers with a higher rate of decline in 32 °C systems during a 6-week operation. Based on the projected residual (observation errors) scatterplot (Figure 5B) where most of the residuals are dispersed and centered around the 0 line, we can assume that the LMEs model indicates proper modeling of the microbiome data

**Figure 5 microorganisms-11-00352-f005:**
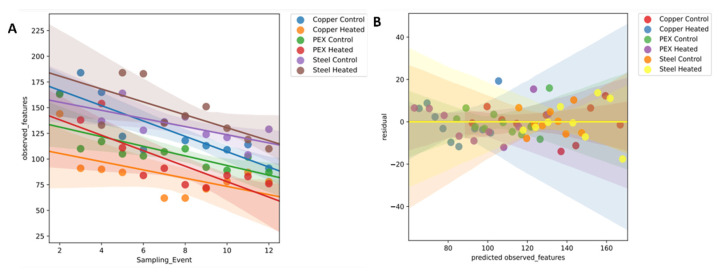
(**A**) Regression scatterplots of LMEs and (**B**) projected residuals plot for each DWDS.

### 3.4. Comparison of the Microbial Community Composition-Beta Diversity

Principal coordinates plot (PCoA) was used to visualize the beta diversity dissimilarity matrices based on Bray–Curtis dissimilarity, Jaccard similarity index, weighted UniFrac, and unweighted UniFrac using an ordination approach adopted from Lozupone, et al. [27]. Pipe materials and temperature conditions both seem to have a major impact on the microbiome composition, however, the temperature condition seems to have a higher magnitude of impact on the bacterial composition. While we noticed a decrease in alpha diversity over time for all the model DWDSs, the beta diversity became more consistent and stable after the first few weeks (Figure 6). A 10 °C increase in water temperature had a significant impact on microbiomes in these systems. Among the three pipe materials, the elevated water temperature had the greatest dissimilarity distance in the PEX system. PERMANOVA analysis showed that each system has significantly different microbiome compositions with a *p*-value ≤ 0.002 and a *pseudo-F* ≥ 4.45 (Table 2). The most pronounced group separation calculated is between 32 °C and 22 °C PEX systems with a pseudo-*F* ratio of 10.928.

**Table 2 microorganisms-11-00352-t002:** Pairwise PERMANOVA comparison of microbial composition (beta diversity) based on pipe material and temperature condition.

Group 1	Group 2	Sample Size	Permutations	Pseudo-F	*p*-Value	q-Value
**Copper 22 °C**	**Copper 32 °C**	19	999	9.696	0.001	0.0012
	**PEX 22 °C**	20	999	4.458	0.001	0.0012
	**PEX 32 °C**	19	999	8.536	0.001	0.0012
	**Steel 22 °C**	20	999	5.699	0.001	0.0012
	**Steel 32 °C**	18	999	5.362	0.001	0.0012
**Copper 32 °C**	**PEX 22 °C**	21	999	7.065	0.001	0.0012
	**PEX 32 °C**	20	999	8.201	0.001	0.0012
	**Steel 22 °C**	21	999	10.648	0.001	0.0012
	**Steel 32 °C**	19	999	10.865	0.001	0.0012
**PEX 22 °C**	**PEX 32 °C**	21	999	10.928	0.001	0.0012
	**Steel 22 °C**	22	999	6.716	0.001	0.0012
	**Steel 32 °C**	20	999	7.977	0.001	0.0012
**PEX 32 °C**	**Steel 22 °C**	21	999	10.651	0.001	0.0012
	**Steel 32 °C**	19	999	7.191	0.002	0.0020
**Steel 22 °C**	**Steel 32 °C**	20	999	5.448	0.002	0.0020

**Figure 6 microorganisms-11-00352-f006:**
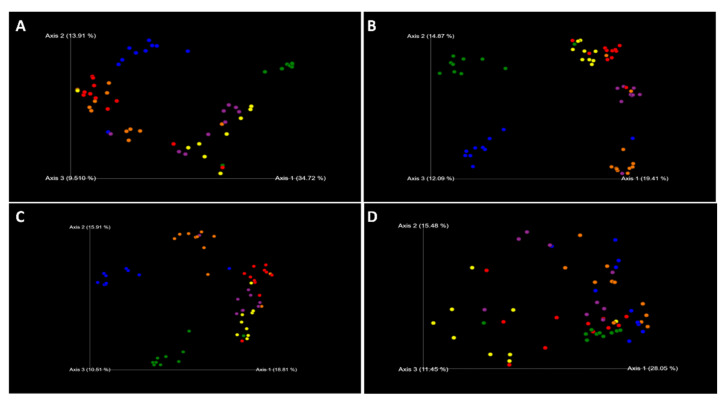
Principal coordinate analysis (PCoA) of the bacterial community compositions in different systems using Bray–Curtis dissimilarity (**A**), Jaccard similarity index (**B**), unweighted UniFrac (**C**), and weighted UniFrac (**D**) NOTE: red: 22 °C copper, blue: 32 °C copper, orange: 22 °C PEX, green: 32 °C PEX, purple: 22 °C steel, yellow: 32 °C steel.

After revelation of some distinct dissimilarity between different model DWDSs in beta diversity metrics and the PERMANOVA analysis, DEICODE was used to predict which taxa are driving the sample clustering [28]. The use of Robust Aitchison PCA via DEICODE revealed clustering in the galvanized steel systems operated at 32 °C and 22 °C (Figure 7).

### 3.5. Prediction of Metagenome Functional Profiles

PICRUSt2 was used to predict the metagenomic functional content of each model DWDS for determining the differentially abundant metabolic pathways between the 32 °C and 22 °C for each pipe material system. PICRUSt2 predicted a total of 328 KEGG pathways in all six model DWDSs. The predicted metabolic pathways that were differentially abundant between 32 °C and 22 °C systems with a statistical significance (*p* < 0.05) are 49, 129 and 35 for copper, PEX and galvanized steel systems, respectively. Only metabolic pathways that have a mean proportion difference of 0.1 or higher and Welch’s test confidence interval (CI) of 0.95 or higher are further investigated (Figure 8). 

### 3.6. Differential Abundance of 32 °C and 22 °C Systems Using ALDEx2

ALDEx2 analysis was applied to identify taxa that significantly differed between the 32 °C and 22 °C systems (Figure 9A). Out of the 555 taxa, analysis of the impact of the temperature condition on the taxa differential abundance revealed 20 taxa that were either differentially abundant on the 32 °C or the 22 °C systems (Figure 9B). The relative abundance and the difference in (clr) of the total 555 taxa are calculated using ALDEx2. From the 20 most significantly differentially abundant taxa (relative abundance, >0.01%), 10 taxa were preferentially found in the 32 °C systems with a significant difference of (>+5 in clr), while the other 10 taxa were more prevalent in the 22 °C systems with a significant difference of (<−5 in clr). Taxa (classified in the Family level) that were differentially abundant in the 32 °C systems are *Comamonadaceae*, *Mycobacteriaceae*, *C__Gemmatimonadetes*, *C__Betaproteobacteria*, *Caulobacteraceae*, *C__Alphaproteobacteria*, *Procabacteriaceae*, *Hyphomonadaceae*, *Rhodospirillaceae*, and *Sinobacteraceae*, while the prevalent taxa in the 22 °C systems are *C__Betaproteobacteria*, *Gemmataceae*, *Hyphomicrobiaceae*, *Caulobacteraceae*, *MLE1-12*(*Cyanobacteria*((*4C0d-2*), *Rhodospirillaceae*, *C__Betaproteobacteria*, *Comamonadaceae*, and *Rhodospirillaceae*. Differentially abundant taxa with unidentified family level were coincidentally not identified on the order level in the GreenGenes ribosomal RNA gene database, therefore, were reported on the class level with (*C__*). Three similar taxa (*Caulobacteraceae, Rhodospirillaceae* and *Comamonadaceae*) were identified in both the 32 °C and 22 °C systems as differentially abundant. Abon further examination of their sequences, the percent identity value is slightly less than 97% which was the threshold used for taxonomy assignments. It is possible that different strains of these taxa (*Caulobacteraceae, Rhodospirillaceae,* and *Comamonadaceae*) are favored in different temperature conditions.

## 4. Discussion

The uniquely designed and operated six-pilot scale model DWDSs used in this study enabled a controlled comparison between the two most important variables, pipe material and water temperature. The study represents a 6-week molecular survey using 16rRNA on different pipe materials and water temperatures. The results indicate that the chemical composition of water was mainly impacted by the type of pipe material, while an increase in water temperature by 10 °C had no impact on the chemistry of water in the model DWDSs. The bacterial composition, on the other hand, was disproportionately impacted by the pipe material and water temperature. Notably, alpha diversity in galvanized steel and copper systems was slightly higher compared to PEX systems, apparently due to their chemical reactivity. The decline in alpha diversity could be due to the relatively rapid proliferation of some bacterial strains in a poor nutrient medium similar to the drinking water environment, however, the total number of species and the rate of decline in each bacterial community seems to be significantly impacted by both pipe material and temperature conditions. Even though fewer strains are dominating PEX systems, the 32 °C and 22 °C PEX systems had the highest dissimilarity distance in beta diversity pairwise comparison. Copper and galvanized steel have greater chemical reactivity compared to PEX, which might explain the higher impact on the microbiome in copper and galvanized steel.

*Proteobacteria* was the most prominent phyla across all systems ranging from 60.9% to 91.1% which has been reported in previous studies [29,30,31]. The frequency of detection of OPs varies depending on the variables such as pipe material and water temperature. Increasing the temperature by 10 °C resulted in a higher prevalence of *Mycobacteriaceae* (100% vs. 58.6%), however, *Pseudomonadaceae* frequency of detection was lower on elevated temperature (53.3% vs. 62.9%). *Legionellaceae* was detected in all the samples of the 32 °C and 22 °C systems. Analysis shows that the beta diversity stabilizes over time despite the consistent decline in alpha diversity. The reduction in alpha diversity over time across all systems despite replenishment with sufficient fresh tap water illustrates the importance of the system’s operational conditions in shaping the DW microbiome. The use of probiotic approaches, as opposed to disinfection, has been suggested to control opportunistic pathogens in the premise plumbing, however, this is one of the areas that require further investigation [32].

The most notable pathways with increased differential abundance in 22 °C compared to 32 °C in copper systems, are the ATP-binding cassette (ABC) transporters, bacterial chemotaxis, bacterial motility proteins, transporters, and two-component system proteins. All these pathways (ABC transporters, bacterial chemotaxis, bacterial motility proteins, and two-component system proteins) have higher abundances in the 22 °C copper system and are known to support bacterial survival and are collectively important for pathogenesis and bacterial responses to the environment [33]. For PEX systems, several metabolic and degradation pathways differed significantly between the 32 °C and 22 °C systems, including beta-alanine metabolism, fatty acid metabolism, lysine degradation, methane metabolism, porphyrin-chlorophyll metabolism, propanoate metabolism, tryptophan metabolism, and valine, leucine, and isoleucine degradation. Methane metabolism and porphyrin-chlorophyll metabolism are the only pathways that have increased differential abundances in the PEX system with elevated temperatures. The reason for the increase in the differential abundance of these two pathways in the 32 °C PEX system could be because of the higher abundances of *Nitrospirales*, known to perform anaerobic methane oxidation coupled with denitrification (DAMO process) [34], and *Gemmatimonadetes*, found to have the ability to perform chlorophyll-based phototrophy [35] and have a strong relationship with methane flux in soil [36]. The galvanized steel systems have the least degree of dissimilarity between the 32 °C and 22 °C with no notable differentially abundant pathways elicited at the two conditions.

The mechanism of interference of pipe material and water temperature on DW microbiome has not been well understood, however, according to Aitchison PCA plot (Figure 7), the galvanized steel systems compositions have very low dissimilarities which suggests that galvanized steel has a significant influence on the DW microbiome. Previous studies have also concluded that galvanized steel pipes tend to have a higher bacterial diversity than any other pipe materials [29,37]. On the other hand, 22 °C copper and PEX systems have comparable diversities which also suggests that water temperature has the potential to alter the bacterial compositions of the DW microbiome. The differential abundance of analysis between the 32 °C and 22 °C systems identified multiple taxa that were present yet differentially abundant between the two systems. Interestingly, *Mycobacteriaceae* was found to be significantly abundant in elevated temperatures, which suggests that there is a correlation between the increase in water temperature and *Mycobacteriaceae. Cyanobacteria* was preferentially found in the 22 °C systems. 

## 5. Conclusions

The study of the DWDSs microbiome has enhanced our understanding of how seasonality, disinfection process and treatment operations impact the microbial community composition [1]. Free-of-microbes DWDSs have been proven an unrealistic approach [32]. Dealing with the drinking water microbiome dynamics is essential in order to develop a probiotic approach for pathogen control in DWDSs in the future. The experimental plan of this work was designed to study the dynamics of the microbiome in six model drinking water distribution systems using three different pipe materials operated at two different temperatures to increase our ability to better predict bacterial species prevalence in drinking water. The results obtained can potentially improve and provide insight into bacterial species detected during municipality monitoring programs. Detailed characterization of microbial data is an essential tool on how to modify operational conditions for minimizing the proliferation of microbial pathogens and improving drinking water quality.

## Figures and Tables

**Figure 1 microorganisms-11-00352-f001:**
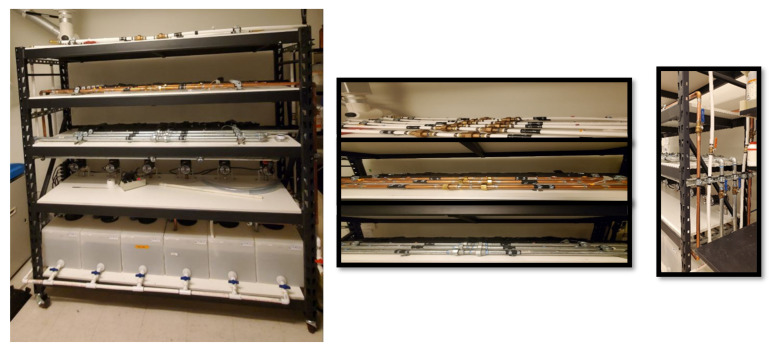
Model drinking water distribution systems—from left to right, front, top and side views. (Front view; tank reservoirs (1st tier, from bottom), pumps (2nd tier), galvanized steel (3rd tier), copper (4th tier), PEX pipes (5th tier)).

**Figure 7 microorganisms-11-00352-f007:**
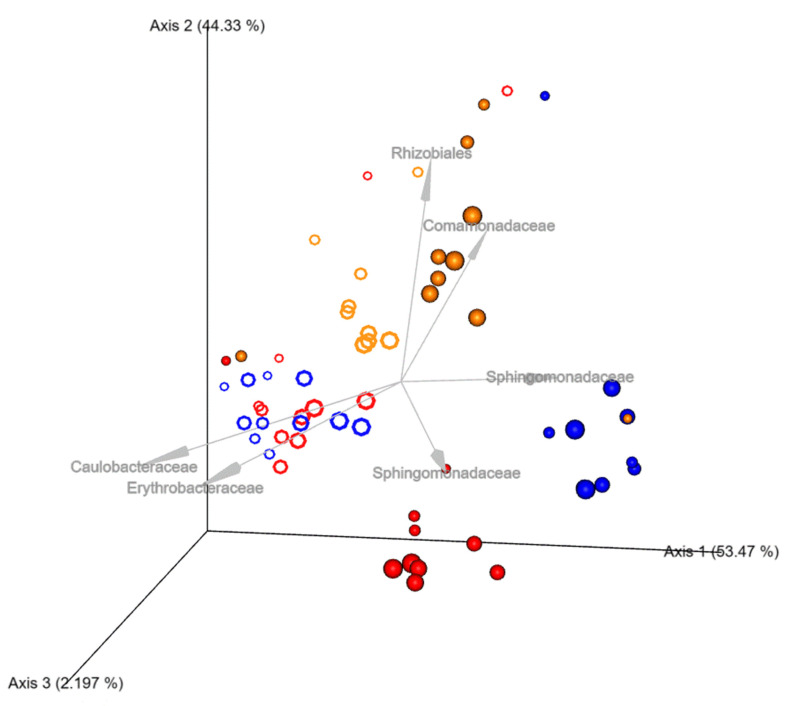
Robust Aitchison PCA generated from DEICODE showing taxa (arrows) correlated with the DWDSs scaled by week, from 1 to 6. NOTE: material code: red: copper, blue: PEX, orange: steel. Temperature code: ○: 22 °C, ●:32 °C.

**Figure 8 microorganisms-11-00352-f008:**
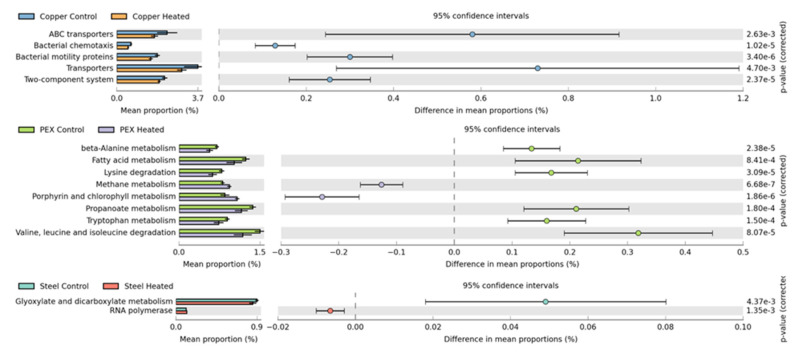
Statistically significant differentially abundant pathways between 32 °C and 22 °C systems.

**Figure 9 microorganisms-11-00352-f009:**
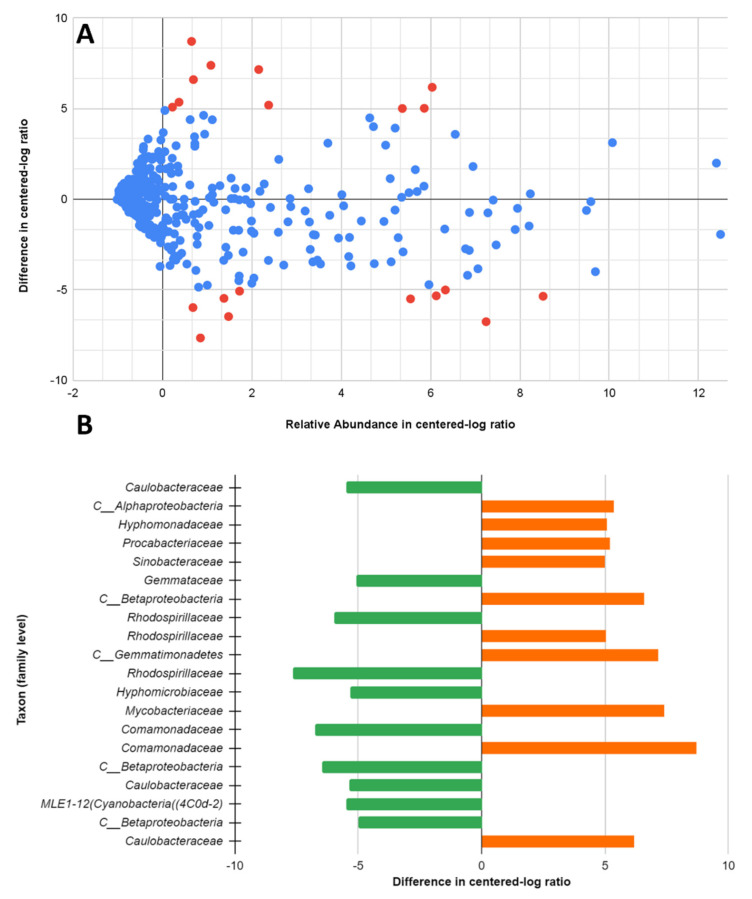
ALDEx2 differential abundance analysis between the 32 °C and 22 °C systems.: (**A**) Bland–Altman or MA plot between relative abundance and difference. (Significantly different taxa are plotted in **red**); (**B**) significantly differentially abundant taxa classified at the family level. (32 °C: green, 22 °C: orange).

## Data Availability

Data from this study are available from the corresponding author upon reasonable request.

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
