# Peer review of "Impact of Pipe Material and Temperature on Drinking Water Microbiome and Prevalence of Legionella, Mycobacterium, and Pseudomonas Species"

_microorganisms, 2023, doi:10.3390/microorganisms11020352_

Round 1

Reviewer 1 Report

This study dealt with an important and interesting topic for studying the effect of selected materials such as drinking water pipes and temperatures on the growth of bacterial flora. The experiment was performed with complete efficiency. If there are any comments, it does not detract from the work.

I have some remarks on the article that may increase the value of the work, as follows:

1. The most dominant organism should be added in the keywords, which is (Legionella, Mycobacterium, and Pseudomonas Species) in order to make it easier for other researchers to access this article when it is published.

2. I have some questions like

a. Does the lengths of these tubes and the speed of water flow play a role in the growth of microorganisms?

b. Does lighting and darkness affect the quality of microorganisms?

c. Does the inside texture of the tubes from the inside, and if there are connections between them, affect the quality and density of microorganisms?

d. Was the sampling time taken into consideration, and was it at close or far intervals? And what is the basis for time of sampling?

3. With regard to Table No. 2, I see that it is placed in the attached data (Supllementary files), not in the manscriupt, and the reference is only to the explanation only in the discussion.

4. I see in the excellent statistical work that was done that if the researchers had used the canonical analyzes (CCA), it would have enriched the work and clarified more.

5. The researchers mention the Evencess and Richness, but not mention in materials and methods.

5. Deletion of any references found in the conglomerate, because this part is concerned with the employers and the extent of their conclusion of their results.

6. The resolution of figures need to be more improved.

Reviewer 2 Report

The main comments about the study.

Knowledge concerning factors which affect microbial quality of drinking water is significant in terms of water safety. The paper underlines that the significance of pipeline material on microbial population structure in drinking water environment.

The major strength of the paper is the general quality of set up of the study and presentation of the results.

Technically the paper is very well written and is easily understandable. The language of the paper follows perfect English grammar. The abstract reflects quite well the content of article. The weakness of the abstract is that it focuses strongly on occurrence of Legionella,  Mycobacteria and Pseudomonas phyla, while the results of the paper are concentrating mainly on general microbial diversity of microbial populations of tested water systems.

The introduction part is quite brief focusing on main factors affecting occurrence of aquatic microbes. The presentation of results is logical and profound. The interpretation of results is done quite well. The interpretation of the results could have contained more references about the similar studies. It can be concluded that the presented conclusions were logically deduced from the results and discussion.

Some minor alterations/modifications can be proposed to improve the paper (see the detailed comments). 

Detailed comments:

Title:

The title indicates that the paper would focus strongly on occurrence of Legionella,  Mycobacteria and Pseudomonas in the tested waters. In fact, the presentation of the results and discussion focuses more on general microbial population structure differences. I propose that the title be changed reflecting more closely the content of the manuscript.

Abstract

Page 1. The weakness of the abstract is that it focuses strongly on occurrence of Legionella,  Mycobacteria and Pseudomonas phyla, while the results are concentrating mainly on general microbial diversity of microbial populations of tested water systems.

Introduction

Pages 1-2. The introduction part is quite concise. There could more about the general effects of pipeline materials, microbial nutrients, temperature, disinfection on biofilm formation  and drinking water quality

Page 1, line 39. Could the authors present a reference concerning microbial counts in water “…103to 106 cells per ml” ?

Page 2, lines 47-48: long distance is one factor, but even more importantly residence time in the pipeline as well affects microbial quality in DWDS.

Material and methods

Page 2, lines 72-73: Could the authors present any information about the variation in incubation temperatures (22áµ’/32áµ’C). Was there temperature monitoring system ? Why temperature 32áµ’C was chosen. Could this temperature be justified ?

Page 2, line 76. Why replenishment of water was done prior the sampling ? Would there be a risk of dilution with fresh water ?

Page 2. Line 77. Water used in the experiments is not determined/described. What kind of water was used – is there any quality information available ?

Page 3, line 86. What was the material, trademark, company of the disk membrane ?

Results

Page 5, line 189. Correct “… Ni and ZN were…” to “… Ni and Zn were…”

Page 7, Figure 4. Larger font size should be used in the titles of the diversity plots (Copper control etc.)

Page 7, lines 253-258. These sentences containing explanations of the results belong to the discussion section.

Page 8, lines 273-274. The sentence should be part of the discussion section.

Page 8, table 2. What is the added value of table 2 in comparison with the main results presented in the page7 (lines 276-279). I propose to omit the table 2.

Discussion

Page 13, lines 382-384: The meaning of the sentence:”… it seems manipulating the environmental factors is more crucial in sustaining the DW microbiome than the introduction of non-naturally existing bacterial consortium.”   is a little bit unclear. Could this be clarified ?

Reviewer 3 Report

This is a comprehensive examination of the microbiome in drinking water and the influence of temperature and and water pipe materials. The use of NGS to study the range of microbial prevalence as well as predominance, supported by chemical analysis, was clearly described.

There were a few minor typos, for example P5 line 189 ZN instead of Zn and P6 L 234 should be capital for Faith's; Fig 2 appears to say -22 rather than 22.

The paper doesn't mention the need and significance of looking at different metabolic pathways within the results section.

It doesn't say what volume of water is sampled for chemical and NGS analysis - was it for example 1 litre which is the minimum standard filtered down for Legionella testing? As it is calculated back to standard units it is not too critical, but interesting I would think to the reader to know what proportion of the system is sampled. 

Obviously only water sampled not biofilm, despite in introduction saying that 95% of biota in biofilm and only 5% in free water - would have been interesting to compare but maybe that's another paper. But some comment could be useful as the microbiome is almost certain to be different in biofilm compared to free water.

My final thought - intuitively before reading the results I expected the microbiome to be smaller / less diverse in the copper pipes just because copper is toxic. For example copper/ silver ions is an established method of legionella control in DWS. So I was surprised it was not the case. But that toxicity is related to free ions - it is not stated in the methods, but were the pipe materials new or established?  The surface of copper will 'passivate' ie acquire a coating that makes it more inert - was that the case? Perhaps some more detail about the test system.  
